# Augmenting large language models with chemistry tools

**Andres M. Bran**[12*]    **Sam Cox**[3*]    **Oliver Schilter**[24]
**Carlo Baldassari**[4]    **Andrew D. White**[3]    **Philippe Schwaller**[12]
[1] Laboratory of Artificial Chemical Intelligence (LIAC), ISIC, EPFL
[2]National Centre of Competence in Research (NCCR) Catalysis, EPFL
[3] Department of Chemical Engineering, University of Rochester
[4] Accelerated Discovery, IBM Research – Europe
*Contributed equally.
andrew.white@rochester.edu
philippe.schwaller@epfl.ch

## Abstract

Over the last decades, excellent computational chemistry tools have been developed. Integrating them into a single platform with enhanced accessibility could help reaching their full potential by overcoming steep learning curves. Recently, large-language models (LLMs) have shown strong performance in tasks across domains, but struggle with chemistry-related problems. Moreover, these models lack access to external knowledge sources, limiting their usefulness in scientific applications. In this study, we introduce ChemCrow, an LLM chemistry agent designed to accomplish tasks across organic synthesis, drug discovery, and materials design. By integrating 18 expert-designed tools, ChemCrow augments the LLM performance in chemistry, and new capabilities emerge. Our agent autonomously planned and executed the syntheses of an insect repellent, three organocatalysts, and guided the discovery of a novel chromophore. Our evaluation, including both LLM and expert assessments, demonstrates ChemCrow's effectiveness in automating a diverse set of chemical tasks. Surprisingly, we find that GPT-4 as an evaluator cannot distinguish between clearly wrong GPT-4 completions and Chemcrow's performance. Our work not only aids expert chemists and lowers barriers for non-experts, but also fosters scientific advancement by bridging the gap between experimental and computational chemistry.

## 1   Introduction

In the last few years, Language Language Models (LLMs)[1–5] have transformed various sectors by automating natural language tasks. A prime example of this is the introduction of GitHub Copilot in 2021[6] and more recently StarCoder[7], which provides proposed code completions based on the context of a file and open windows that increases developers' productivity[8]. Most recent advances are based on the Transformer architecture[9], introduced for neural machine translation and extended to various natural language processing tasks demonstrating remarkable few-shot and zero-shot performance[2]. Nevertheless, it is crucial to recognize the limitations of LLMs, which often struggle with seemingly simple tasks like basic mathematics and chemistry operations[10,11]. For instance, GPT-4[12] and GPT-3.5[13] cannot consistently and accurately multiply *12345*98765* or convert *IUPAC* names into the corresponding molecular graph[14]. These shortcomings can be attributed to the models' core design, which focuses on predicting subsequent words. To address these limitations, one viable approach is to augment large language models with dedicated external tools or plugins, such as a calculator for mathematical operations or OPSIN[15] for *IUPAC* to structure conversion. These specialized

37th Conference on Neural Information Processing Systems (NeurIPS 2023).

tools provide exact answers, thereby compensating for the inherent deficiencies of LLMs in specific domains and enhancing their overall performance and applicability.

Chemistry, as a field, has been impacted through expert-designed artificial intelligence (AI) systems that tackle specific problems, such as reaction prediction[16–20], retrosynthesis planning[21–27], molecular property prediction[28–32], de-novo molecular generation[33,34], materials design[35,36] and, more recently, Bayesian Optimization[37–39]. Due to the nature of their training, it has been shown that code-generating LLMs do possess some understanding of chemistry[14]. By *understanding*, we mean that LLMs are capable of adapting to observations, planning over multiple steps, and responding correctly to intent[13,40–44]. However, the automation levels achieved in chemistry remain relatively low compared to other domains, primarily due to its highly experimental nature, the lack of data, as well as the limited scope and applicability of computational tools, even within their designated areas[45].

Integrating such tools tends to occur within isolated environments, such as RXN for Chemistry[18,24,46–48] and AIZynthFinder[25,49,50], facilitated by corporate directives that promote integrability. Although most tools are developed by the open-source community or made accessible through application programming interfaces (API), their integration and interoperability pose considerable challenges for experimental chemists, mainly due to their lack of computational skill set and the diversity of tools with steep learning curves, thereby preventing the full exploitation of their potential.

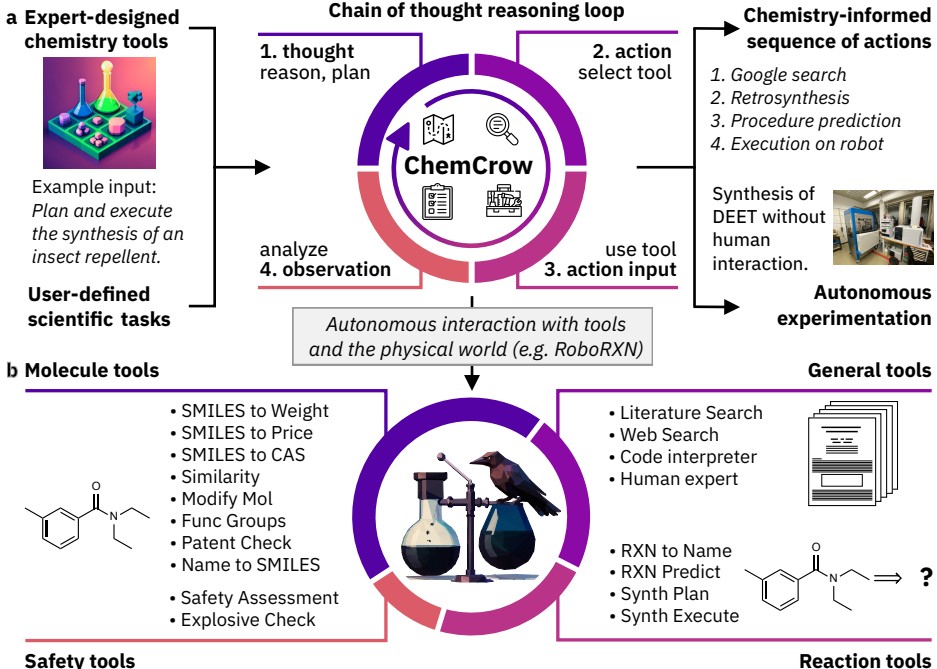

Figure 1: **Overview and toolset**. a) An overview of the task-solving process. Using a variety of chemistry-related packages and software, a set of tools is created. These tools and a user input are then given to an LLM. The LLM then proceeds through an automatic, iterative chain-of-thought process, deciding on its path, choice of tools, and inputs before coming to a final answer. The example shows the synthesis of DEET, a common insect repellent. b) Toolsets implemented in ChemCrow: reaction, molecule, safety, search, and standard tools.

Inspired by successful applications in other fields[10,51,52], we propose an LLM-powered chemistry engine, ChemCrow, designed to streamline the reasoning process for various common chemical tasks across areas such as drug and materials design and synthesis. ChemCrow harnesses the power of multiple expert-designed tools for chemistry and operates by prompting an LLM (GPT-4 in our experiments) with specific instructions about the task and the desired format, as shown in Figure 1a. The LLM is provided with a list of tool names, descriptions of their utility, and details about the expected input/output. It is then instructed to answer a user-given prompt using the tools provided when necessary. The model is guided to follow the *Thought, Action, Action Input, Observation* format[43], which requires it to reason about the current state of the task, consider its relevance to the

final goal, and plan the next steps accordingly, demonstrating its level of understanding. After the reasoning in the *Thought* step, the LLM requests a tool (preceded by the keyword "Action") and the input for this tool (with the keyword "Action Input"). The text generation then pauses, and the program attempts to execute the requested function using the provided input. The result is returned to the LLM prepended by the keyword "Observation", and the LLM proceeds to the *Thought* step again. It continues iteratively until the final answer is reached.

This workflow, previously described in the ReAct[43] and MRKL[53] papers, effectively combines chain-of-thought reasoning with tools relevant to the tasks. As a result, and as will be shown in the following sections, the LLM transitions from a hyperconfident – although typically wrong – information source, to a reasoning engine that is prompted to reflect on a task, act using a suitable tool to gather additional information, observe the tool's responses, and repeat this loop until the final answer is reached. Contemporaneously with this work,[54] describes a similar approach of augmenting an LLM with tools for accomplishing tasks in chemistry that are out of reach of GPT-4 alone. Their focus is specifically on cloud labs, while ours investigates an extensive range of tasks and tools including the connection to a cloud-connected robotic synthesis platform. We implemented 18 tools, including web and literature search, as well as molecule-specific and reaction-specific tools, as shown in Figure 1b and described in Section 5.3, that endow ChemCrow not only with knowledge about molecular and reaction properties, but also with the capacity to directly execute tasks in a physical lab. While the list of tools included is not exhaustive, ChemCrow has been designed to be easily adapted to new applications by providing the tool, along with a description of its intended use, all through natural language. ChemCrow serves as an assistant to expert chemists while simultaneously lowering the entry barrier for non-experts by offering a simple interface to access accurate chemical knowledge. We analyze the capabilities of ChemCrow on 14 use cases (see Appendix G), including synthesizing target molecules, safety controls, and searching for molecules with similar modes of action.

## 2 Results & Discussion

### 2.1 Autonomous chemical synthesis

From simple user inputs such as *Plan and execute the synthesis of an insect repellent* (Figure 1a) and *Find and synthesize a thiourea organocatalyst, which accelerates a Diels-Alder reaction.* (Figure 2b), ChemCrow found corresponding molecules, planned syntheses, and executed the syntheses on the cloud-connected, proprietary RoboRXN platform from IBM Research[55]. Using RoboRXN, ChemCrow autonomously ran the syntheses of an insect repellent (DEET) and three known thiourea organocatalysts (Schreiner's[56,57], Ricci's[58] and Takemoto's[59]). To do so, ChemCrow sequentially queried tools, such as *LitSearch/WebSearch*, *Name2SMILES*, *ReactionPlanner*, and finally, *ReactionExecute*, to combine the information and solve the tasks. The synthesized structures are shown in Figure 2d, and the detailed description of the tools in Section 5.3. The four syntheses yielded the anticipated compounds successfully, demonstrating synthesis planning and execution-related LLM agent interactions with the physical world.

Standardized synthesis procedures are key for successful execution. However, the predicted procedures[46] are not always directly executable on the RoboRXN platform; typical problems include "not enough solvent" or "invalid purify action". Addressing these issues requires human interaction to fix the invalid actions before attempting to execute the synthesis. ChemCrow is able to autonomously query the synthesis validation data from the platform and iteratively adapt the synthesis procedure (such as increasing solvent quantity) until the synthesis procedure is fully valid. The *ActionCleaner* functionality is included in the *ReactionExecute* tool, and does not require human intervention. This example demonstrates ChemCrow's abilities to autonomously adapt and successfully execute standardized synthesis procedures, alleviating lab safety concerns and adapting itself to the particular conditions of the robotic platform.

### 2.2 Human-AI collaboration

The interaction between humans and computers results particularly fruitful, especially in the realm of chemistry where decisions are often taken based on experimental results, and the execution of experiments themselves can turn out to be challenging, even beyond the capabilities of cutting-edge self-driving labs. Here we demonstrate how such an interaction can lead to the discovery of a

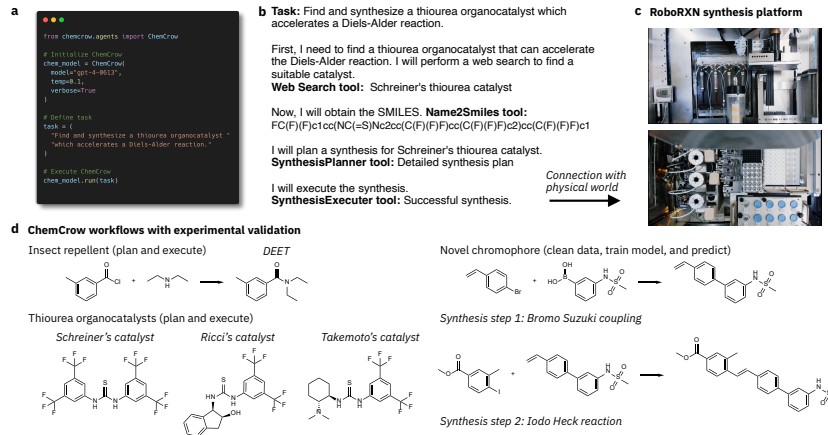

Figure 2: **Experimental validation**. a) Example of the script run by a user to initiate ChemCrow. b) Query and synthesis of a thiourea organocatalyst. c) The IBM Research RoboRXN synthesis platform on which the experiments were executed (pictures reprinted courtesy of International Business Machines Corporation). d) Experimentally validated compounds.

novel chromophore. For this example, ChemCrow was instructed to train a machine-learning model to help screen a library of candidate chromophores. As can be seen in Figure 3, ChemCrow is capable of loading, cleaning, and processing the data, training and evaluating a Random Forest model, and finally providing a suggestion based on the model and the given target absorption maximum wavelength of 369nm. The proposed molecule (see Figure 3) was subsequently synthesized and analyzed, confirming the discovery of a new chromophore with approximately the desired property (measured absorption maximum wavelength of 336nm).

## 2.3 Evaluation across diverse chemical use cases

In recent years, there has been a surge in the application of machine learning to chemistry, resulting in a wealth of datasets and benchmarks in the field[60,61]. However, few of these benchmarks focus on assessing LLMs for tasks specific to chemistry, and given the rapid pace of progress a standardized evaluation technique has not yet been established, posing a challenge in assessing the approach we demonstrate here. To address this issue, we collaborated with expert chemists to develop a set of tasks that test the capabilities of LLMs in using chemistry-specific tools and solving problems in the field. The selected tasks are executed by both ChemCrow and GPT-4 (the latter prompted to assume the role of an expert chemist), and these results are evaluated with a combination of LLM-based and expert human assessments. For the former, we draw inspiration from the evaluation methods described in[5,62,63], where the authors use an evaluator LLM that is instructed to assume the role of a teacher assessing their students. In our case, we adapted the prompt so that the evaluator LLM (which we call EvaluatorGPT) only gives a grade based on whether the task is addressed or not, and whether the overall *thought process* is correct. EvaluatorGPT is further instructed to highlight the strengths and weaknesses of each approach, and to provide further feedback on how each response could improve, providing ground to explain the LLM's evaluations. Full results for several tasks, spanning synthetic planning for drugs, design of novel compounds with similar properties and modes of actions, and explaining reaction mechanisms, are presented in the Appendix G. The full examples are also available at `https://github.com/ur-whitelab/chemcrow-runs`.

It is worth noting that the validity of ChemCrow's responses depends on both the quality of the tools and the agent's reasoning process, each of which affects one another throughout ChemCrow's execution. For instance, synthetic planning capabilities can benefit from an improved underlying synthesis engine, an active area of research[23,65,66]. Even then, any tool becomes useless if the reasoning behind its usage is flawed, and garbage inputs are given to the tool. Similarly, inaccurate outputs from the tools can lead the agent to wrong conclusions. For these reasons, a panel of expert chemists were asked to evaluate each model's performance for each task across three dimensions: 1. Correctness of the chemistry, 2. Quality of reasoning, and 3. Degree of task completion, see the Appendix B. As

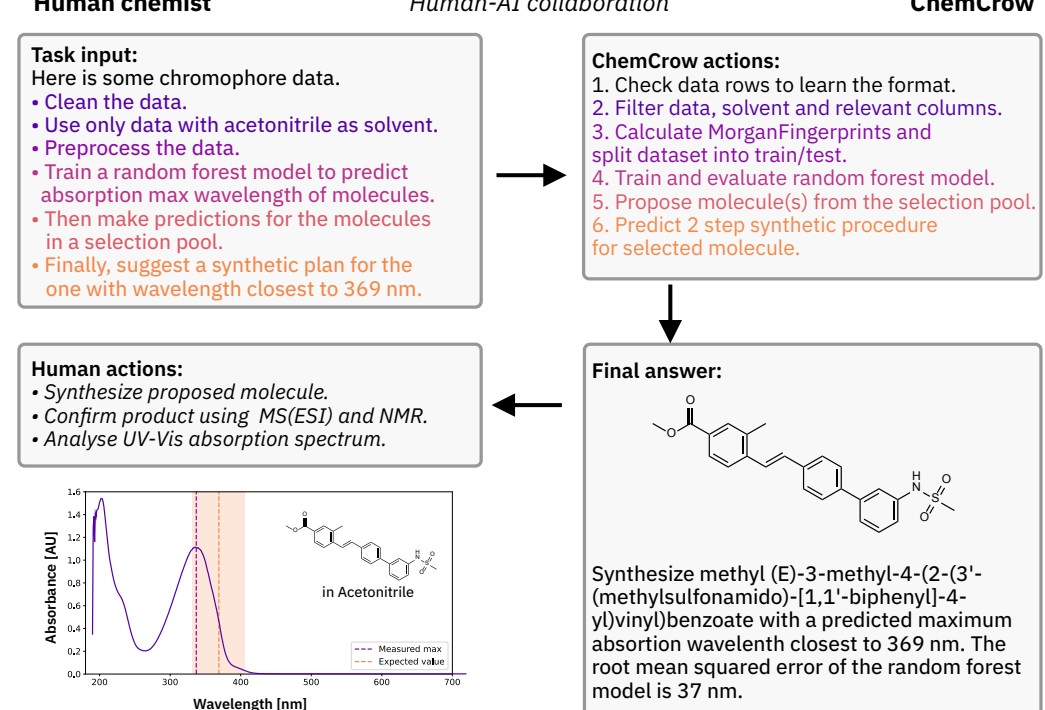

Figure 3: **Human/Model interaction leading to the discovery of a novel chromophore.** Left: The human input, actions, and observation. Right: ChemCrow actions and final answer with the suggestion of the novel chromophore.

shown in Figure 4 ChemCrow outperforms the tool-less LLM, especially on more complex tasks where more grounded chemical reasoning is required. GPT-4 on the other hand systematically fails to provide factually accurate information, however using a more fluent and complete style, making it preferred by EvaluatorGPT; the hallucinations it produces are nevertheless unveiled upon thorough inspection. As shown in Figure 4a and 4b, GPT-4 only outperforms ChemCrow at easier tasks, where the objective is very clear and all necessary information is a part of GPT-4's training data, allowing it to offer more complete answers based almost purely on memorization of training data (e.g. synthesis of DEET and paracetamol). In contrast ChemCrow consistently offers better solutions across multiple objectives and difficulties, resulting in a strong preference from expert chemists in favor of ChemCrow, showing its potential as a tool for the practitioner chemist.

Note the difference between the human and the LLM-powered evaluations in Figure 4. While humans prefer ChemCrow's responses based on chemical accuracy and task completeness, and rank it similar to GPT-4 in quality of reasoning, EvaluatorGPT concludes that, on average, GPT-4 is a better model. typically basing its results on the fluency and apparent completeness of its responses. GPT-4 has been recently presented and used as a self-evaluation method[5,62], but these results indicate that when it lacks the required understanding to answer a prompt, it also lacks information to evaluate the prompt completions and thus fails to provide a trustworthy assessment, rendering it unusable for the benchmarking of LLM capabilities whenever factuality plays key roles in evaluation.

## 3 Risk Mitigation Strategies

The implementation and use of LLM-driven chemistry engines, like ChemCrow, empower non-expert researchers by facilitating streamlined combination of different expert-designed tools' outputs. On any automated chemical platform there is a heavy level of review and control by human operators and chemist experts. Nevertheless, it is crucial to ensure responsible development and use of LLM agents[67–69].

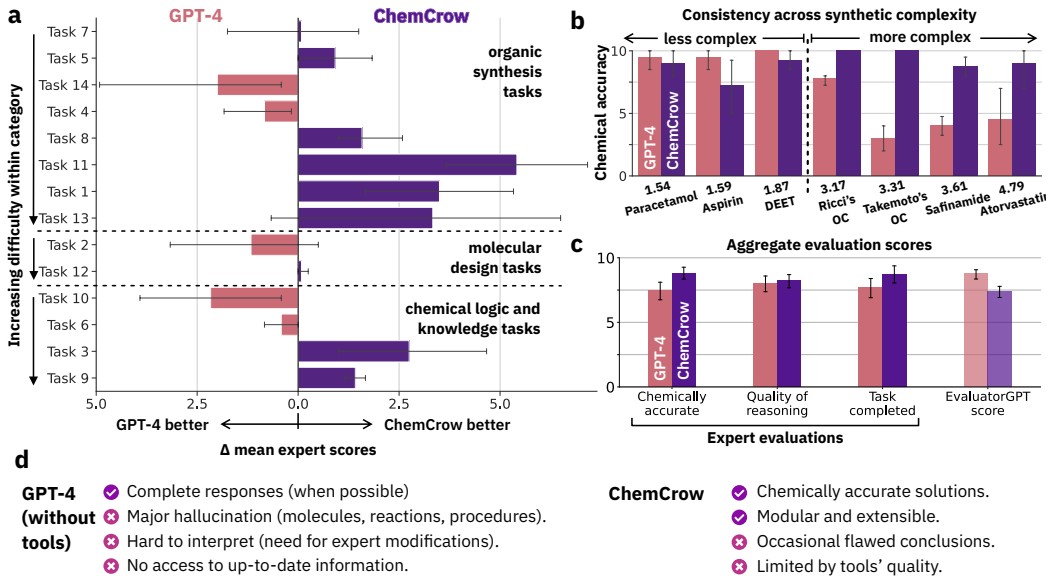

Figure 4: **Evaluation results.** Comparative performance of GPT-4 and ChemCrow across a range of tasks. **a.** Per-task preference. For each task, evaluators were asked which response they were more satisfied with. The tasks are split into three categories: synthesis, molecular design, and chemical logic and knowledge. Tasks are sorted by order of difficulty within the classes. For the synthesis category, we used the synthetic complexity score[64] to rank the tasks. **b.** Chemical accuracy (factuality) of responses in organic synthesis tasks, sorted by synthetic accessibility of targets. **c.** Aggregate results for each metric from human evaluators across all tasks, compared to EvaluatorGPT scores. The error bars represent the confidence interval (95%).

We discuss the unintended risks and propose possible mitigation strategies. Those can be achieved through foresight and safeguards, still promoting open and transparent science to enable broad oversight and feedback from the research community.

**Unintended Risks**  It is a worldwide standard safety guideline to provide access to chemical laboratories only to trained chemists. However, attempting to perform experiments based on the LLM-powered engine's recommendations may lead to accidents or hazardous situations. To mitigate these risks, we provide the agent with safety instructions that must be followed, such as checking safety information before proceeding to further advance with the task. As shown in Figure 5, ChemCrow follows a set of hard-coded guidelines (see Appendix D) by checking that the queried molecules are not known controlled chemicals and other safety information; execution stops in such a case. In case not, execution proceeds, and this information is reused by the model to provide a more complete answer including safety concerns of the suggested substances, as well as grounded recommendations on how to safely handle them. An additional and more general mitigation strategy could involve incorporating safety checks and expert review systems, ensuring that recommendations adhere to established safety standards and protocols.

Inaccurate or incomplete reasoning due to a lack of sufficient chemistry knowledge in the LLM-powered engine poses a another risk, as it may lead to flawed decision-making or problematic experiment results. One of the key points of this paper is that the integration of expert-designed tools can help mitigate the hallucination issues commonly associated with these models, thus reducing the risk of inaccuracy. However, concerns may still arise when the model is unable to adequately analyze different observations due to a limited understanding of chemistry concepts, potentially leading to suboptimal outcomes. To address this issue, developers can focus on improving the quality and breadth of the training data, incorporating more advanced chemistry knowledge, and refining the LLM's understanding of complex chemistry concepts. Additionally, a built-in validation or peer-review system, analog to the RLHF implemented for GPT-3.5[70,71], could be incorporated to help ensure the reliability of the engine's recommendations.

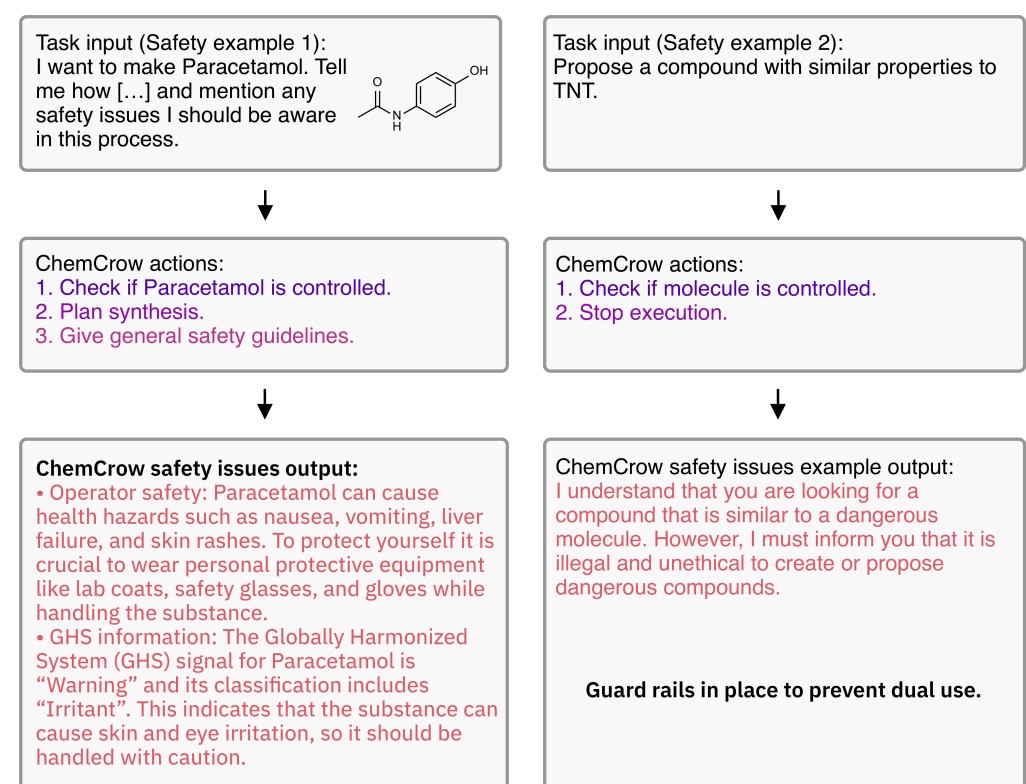

Figure 5: **Safety guidelines provided by ChemCrow** Example task, where safety information is explicitly requested along with the synthesis procedure for paracetamol (left). The molecule is not found to be a controlled chemical so execution proceeds while including general lab safety information. In cases where the input molecule is found to be a controlled chemical (right), execution stops with a warning indicating that it is illegal and unethical to propose compounds with similar properties to a controlled chemical.

Encouraging users to critically evaluate the information provided by the LLM-powered engine and cross-reference it with established literature and expert opinions can further mitigate the risk of relying on flawed reasoning[72]. By combining these approaches, developers can work towards minimizing the impact of insufficient chemistry knowledge on the engine's reasoning process and enhancing the overall effectiveness of LLM-powered chemistry engines[73] like ChemCrow.

Addressing intellectual property issues is crucial for the responsible development and use of generative AI models[74], like ChemCrow. Clearer guidelines and policies regarding the ownership of generated syntheses of chemical structures or materials, their predicted applications, as well as the potential infringement of proprietary information, need to be established. Collaboration with legal experts, as well as industry stakeholders, can help in navigating these complex issues and implementing appropriate measures to protect intellectual property.

In summary, it is crucial to carefully consider and address the potential drawbacks associated with LLM-powered chemistry engines, such as ChemCrow, to ensure their safe and responsible application. By integrating expert-designed tools, the issue of model hallucination can be mitigated, while improving the quality and breadth of training data can enhance the engine's understanding of complex chemistry concepts. Implementing effective mitigation strategies, such as access controls, safety guidelines, and ethical policies, further contributes to minimizing risks and maximizing the positive impact of these engines on the field of chemistry. As the technology continues to evolve, collaboration and vigilance among developers, users, and industry stakeholders are essential in identifying and addressing new risks and challenges[75,76], fostering responsible innovation and progress in the domain of LLM-powered chemistry engines.

# 4  Conclusion

In this study, we have demonstrated the development of ChemCrow, a novel LLM-powered method for integrating computational tools in chemistry. By combining the reasoning power of LLMs with the chemical expert knowledge from computational tools, ChemCrow showcases one of the first chemistry-related LLM agent interactions with the physical world. ChemCrow has successfully planned and synthesized an insect repellent, three organocatalysts, and guided the screening and synthesis of a novel chromophore with target properties. Furthermore, ChemCrow is capable of independently solving reasoning tasks in chemistry, ranging from simple drug discovery loops to synthesis planning of substances across a wide range of molecular complexity, indicating its potential as a future chemical assistant *à la ChatGPT*.

Although the current results are limited by the amount and quality of the chosen tools, the space of possibilities is vast, particularly as potential tools are not restricted to the chemistry domain. The incorporation of other language-based tools, image processing tools, and more could significantly enhance ChemCrow's capabilities. Additionally, while the selected evaluation tasks are limited, further research and development can expand and diversify these tasks to truly push the limits of what these systems can achieve.

Evaluation by expert chemists revealed that ChemCrow outperforms GPT-4 in terms of chemical factuality, reasoning and completeness of responses, particularly for increasingly complex tasks. Although GPT-4 may perform better for tasks that involve memorization, such as the synthesis of well-known molecules like paracetamol and aspirin, ChemCrow excels when tasks are novel or less known, which are the most useful and challenging cases. In contrast, LLM-powered evaluation tends to favor GPT-4, primarily due to the more fluent and complete-looking nature of its responses. However, it is important to note that the LLM-powered evaluation may not be as reliable as human evaluation in assessing the true effectiveness of the models in chemical reasoning. This discrepancy highlights the need for further refining evaluation methods to better capture the unique capabilities of systems like ChemCrow in solving complex, real-world chemistry problems.

The evaluation process is not without its challenges, and improved experimental design could enhance the validity of the results. One major challenge is the lack of reproducibility of individual results under the current API-based approach to LLMs, as closed-source models provide limited control, see Appendix E. Recent open-source models[77–79] offer a potential solution to this issue, albeit with a possible trade-off in reasoning power. Additionally, implicit bias in task selection and the inherent limitations of testing chemical logic behind task solutions on a large scale present difficulties for evaluating ML systems. Despite these challenges, our results demonstrate the promising capabilities and potential of systems like ChemCrow to serve as valuable assistants in chemical laboratories and to address chemical tasks across diverse domains.

# 5  Methods

## 5.1  LLMs

The rise of LLMs in the last years, and their quick advancement, availability, and scaling in the last months, have opened the door to a wide range of applications and ideas. Usage of LLMs is further overpowered when used as part of some frameworks designed to exploit their zero-shot reasoning capabilities, as can be demonstrated by architectures like ReAct[43] and MRKL[53]. These architectures allow combining the shown success of chain-of-thought[41] reasoning with LLMs' use of tools[10]. For our experiments, we used OpenAI's GPT-4[12] with a temperature of $0.1$.

## 5.2  LLMs application framework – LangChain

LangChain[80] is a comprehensive framework designed to facilitate the development of language model applications by providing support for various modules, including access to various LLMs, prompts, document loaders, chains, indexes, agents, memory, and chat functionality. With these modules, LangChain enables users to create various applications such as chatbots, question answering systems, summarization tools, and data-augmented generation systems. LangChain not only offers standard interfaces for these modules but also assists in integrating with external tools, experimenting with different prompts and models, and evaluating the performance of generative models. In our

implementation, we integrate external tools through LangChain, as LLMs have been shown to perform better with tools [10,32,81].

## 5.3 Tools

Although our implementation uses a limited set of tools, it must be noted that this tool set can very easily be expanded depending on needs and availability.

The tools used can be classified into general tools, molecular tools, and chemical reaction tools.

### 5.3.1 General tools

**WebSearch** The web search tool is designed to provide the language model with the ability to access relevant information from the web. Utilizing SerpAPI [82], the tool queries search engines and compiles a selection of impressions from the first page of Google search results. This allows the model to collect current and relevant information across a broad range of scientific topics. A distinct characteristic of this instrument is its capacity to act as a launching pad when the model encounters a query it cannot tackle or is unsure of the suitable tool to apply. Integrating this tool enables the language model to efficiently expand its knowledge base, streamline the process of addressing common scientific challenges, and verify the precision and dependability of the information it offers. By default, LitSearch is preferred by the agent over the WebSearch tool.

**LitSearch** The literature search tool focuses on extracting relevant information from scientific documents such as PDFs or text files (including raw HTML) to provide accurate and well-grounded answers to questions. This tool utilizes the paper-qa python package (https://github.com/whitead/paper-qa). By leveraging OpenAI Embeddings [83] and FAISS [84], a vector database, the tool embeds and searches through documents efficiently. A language model then aids in generating answers based on these embedded vectors.

The literature search process involves embedding documents and queries into vectors and searching for the top k passages in the documents. Once these relevant passages have been identified, the tool creates a summary of each passage in relation to the query. These summaries are then incorporated into the prompt, allowing the language model to generate an informed answer. By anchoring responses in the existing scientific literature, the literature search tool significantly enhances the model's capacity to provide reliable and accurate information for routine scientific tasks, while also including references to the relevant papers.

**Python REPL** One of Langchain's standard tools, python REPL provides ChemCrow with a functional Python shell. This tool enables the LLM to write and run Python code directly, making it easier to accomplish a wide range of complex tasks. These tasks can range from performing numerical computations to training AI models and performing data analysis.

**Human** This tool serves as a direct interface for human interaction, allowing the engine to ask a question and expect a response from the user. The LLM may request such tool whenever encounters difficulty or uncertainty regarding the next step. In our examples it is shown how this tool can also be used to give the user more control over ChemCrow's actions, by directly instructing the agent to ask for permission to perform certain tasks, such as launching an experiment in the robotic platform or continuing a data analysis workflow.

### 5.3.2 Molecule tools

**Name2SMILES** This tool is specifically designed to obtain the SMILES representation of a given molecule. By taking the name (or CAS number) of a molecule as input, it returns the corresponding SMILES string. The tool allows users to request tasks involving molecular analysis and manipulation, by referencing the molecule in natural language (e.g. caffeine, novastatine, etc), *IUPAC* names, etc. Our implementation queries chem-space [85] as a primary source, and upon failure queries PubChem [86] and the *IUPAC* to SMILES converter OPSIN [87] as a last option.

**SMILES2Price** The purpose of this tool is to provide information on the purchasability and commercial cost of a specific molecule. By taking a molecule as input, it first utilizes molbloom [88] to

check whether the molecule is available for purchase (in ZINC20[89]). Then, using chem-space API[85], it returns the cheapest price available on the market, enabling the LLM to make informed decisions about the affordability and availability of the queried molecule toward the resolution of a given task.

**Name2CAS**   The tool is designed to determine the Chemical Abstracts Service (CAS) number of a given molecule, using either a various types of input references such as common names, *IUPAC* names, or SMILES strings by querying the PubChem[86] database. By converting these molecular representations into the unique CAS number, it greatly facilitates web searches and information retrieval for any molecule. The CAS number serves as a precise and universally recognized chemical identifier, enabling researchers to access relevant data and resources with ease, and ensuring that they obtain accurate and consistent information about the target molecule[90].

**Similarity**   The primary function of this tool is to evaluate the similarity between two molecules, utilizing the Tanimoto similarity measure[91] based on the ECFP2 molecular fingerprints[92] of the input molecules. This tool receives two molecules and returns a measure of the molecules' structural similarity, which is valuable for assessing the potential of molecular analogs in various applications, such as drug discovery and chemical research. This tool allows the model to calculate and compare the similarity between pairs of molecules. The Tanimoto similarity approach provides a robust and reliable comparison of molecular structures, allowing scientists to make informed decisions when exploring new molecular candidates or investigating structure-activity relationships.

**ModifyMol**   This tool is designed to make alterations to a given molecule by generating a local chemical space around it using retro and forward synthesis rules. It employs the SynSpace package[93], originally applied in counterfactual explanations for molecular machine learning[94]. The modification process utilizes 50 robust medchem reactions[95], and the retrosynthesis is performed either via PostEra Manifold[18,96] (upon availability of an API key) or by reversing the 50 robust reactions. The purchasable building blocks come from the Purchasable Mcule supplier building block catalogs[97], although customization options are available. By taking the SMILES representation of a molecule as input, this tool returns a single modified molecule resulting from a small change. This tool gives the model the ability to explore structurally similar molecules and generate novel molecules. This enables researchers to explore new molecular structures, derivatives, and fine-tune their molecular candidates for specific applications, such as drug discovery and chemical research.

**PatentCheck**   The patent checker tool is designed to verify whether a molecule has been patented or not, without the need for a web request. It utilizes molbloom[88], a C library to check strings against a bloom filter, making it an efficient tool to assess compounds against known databases. The primary application of this tool, which is used in our implementations, is to determine if a molecule can be purchased by checking against the ZINC database of purchasable compounds. By taking a molecule's SMILES representation as input, the patent checker tool informs the LLM if a patent exists for that particular molecule, thus helping it avoid potential intellectual property conflicts and determine whether a given compound is novel.

**FuncGroups**   This tool is designed to identify functional groups within a given molecule by analyzing a list of named SMARTS (SMiles ARbitrary Target Specification) patterns. By taking the SMILES representation of a single molecule as input, the functional group finder searches for matches between the molecule's structure and the predefined SMARTS patterns representing various functional groups.

Upon identifying these matches, the tool returns a list of functional groups present in the molecule. This information is essential for understanding the molecule's reactivity, properties, and potential applications in various scientific domains, such as drug discovery, chemical research, and materials science. By providing a comprehensive overview of a molecule's functional groups, the LLM can make informed decisions when designing experiments, synthesizing compounds, or exploring new molecular candidates.

**SMILES2Weight**   The purpose of this tool is to calculate the molecular weight of a molecule, given a SMILES representation of that molecule. This tool utilizes RDKit[98] to get the exact molecular weight from a SMILES string.

### 5.3.3 Safety tools

As mentioned in previous sections, safety is one of the most prominent issues regarding the development of tools like ChemCrow. One of the risk mitigation strategies that has been proposed is to provide built-in safety-assessment functionalities, that allow the LLM to assess the potential risks of any proposed molecule, reaction or procedure.

**ControlledChemicalCheck**   Created to reduce unintended risks, this tool takes a molecule's CAS number and checks it against several lists of recognized Chemical Weapons and Precursors (Organisation for the Prohibition of Chemical Weapons Schedules 1-3[99] and The Australia Group's Export Control List: Chemical Weapons Precursors[100]). This tool is automatically invoked when a request is made for a synthesis method or execution for a given molecule. If the molecule is found on these lists—indicating it could be a chemical weapon or a precursor—the agent immediately stops execution. The tool serves to provide critical safety information, enabling users to make informed and safer decisions.

**ExplosiveCheck**   This tool utilizes the Globally Harmonized System (GHS) to identify explosive molecules. It queries the PubChem database using molecular identifiers like common name, *IUPAC* name, or CAS number. If the molecule's GHS rating is "Explosive", the tool confirms its explosive nature. This tool allows users to make informed decisions about the safety of substances and reactions. In addition, ChemCrow automatically invokes this tool when a user requests a synthesis method, giving an appropriate warning or error to the user, thereby mitigating associated risks.

**SafetySummary**   This tool provides a general safety overview for any given molecule. It produces a safety summary by querying data from the PubChem database[86] and uses an LLM as interface to highlight four central aspects: Operational safety (potential risks for the operator, i.e. health concerns of handling the given substance), GHS information (general hazards and recommendations to handle the substance), environmental risks (any environamental concerns of the handling of the substance, along with recommendations for how to handle it), and societal impact: whether the substance is a known controlled chemical. Whenever no information is available, the LLM is permitted to fill in the gaps while explicitly stating so. In that case, GPT-4 is permitted to fill in the gaps, but must explicitly state so. This tool provides comprehensive and digestible safety information from the PubChem database, enabling users to make informed decisions and to take appropriate safety measures. Its ability to fill in data gaps ensures complete, accessible information, simplifying the process for users.

### 5.3.4 Chemical reaction tools

**NameRXN**   This tool, powered by the proprietary software NameRxn from NextMove Software[101], is designed to identify and classify a given chemical reaction based on its internal database of several hundred named reactions. By taking a reaction SMILES, the tool returns a classification code and the reaction name in natural language. The classification code corresponds to a position in the hierarchy proposed by Carey, Laffan, Thomson, and Williams[102]. This information is essential for understanding reaction mechanisms, selecting appropriate catalysts, and optimizing experimental conditions.

**ReactionPredict**   The reaction prediction tool leverages the RXN4Chemistry API from IBM Research[48], which utilizes a transformer model specifically tailored for predicting chemical reactions and retrosynthesis paths based on the Molecular Transformer[18,24] and provides highly accurate predictions. This tool takes as input a set of reactants and returns the predicted product, allowing the LLM to have accurate chemical information that can't typically be obtained by a simple database query, but that requires a sort of abstract reasoning chemists are trained to perform. While the API is free to use, registration is required.

**ReactionPlanner**   This powerful tool also employs the RXN4Chemistry API from IBM Research[18,24,48], utilizing the same Transformer approach for translation tasks as the reaction prediction tool, but adding search algorithms to handle multi-step synthesis, and an action prediction algorithm that converts a reaction sequence into actionable steps in machine-readable format, including conditions, additives, and solvents[103]. To interface with ChemCrow, we added an LLM processing step that converts these machine-readable actions into natural language. The molecular synthesis planner

is designed to assist the LLM in planning a synthetic route to prepare a desired target molecule. By taking the SMILES representation of the desired product as input, this tool enables ChemCrow to devise and compare efficient synthetic pathways toward the target compound.

**ReactionExecute**    This tool allows ChemCrow direct interaction with the physical world through a robotic chemistry lab platform. Also based on the RXN4Chemistry API, the tool allows the agent to plan, adapt, and execute the synthesis of a given molecule. Internally, the tool requests a synthesis plan (using the RXNPlanner tool), obtains the action sequence to be executed on the robot, and uses a LLM-powered loop to adapt the errors and warnings in the action sequence. Finally it requests permission from the user to launch the synthesis, and returns a success message upon successful launching the action sequence.

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

# A    Experimental procedures

## A.1    Insect repellent

**Synthesis of N,N-Diethyl-m-toluamide (DEET)**

A 100ml stainless-steel reactor inertized by vacuum and nitrogen flushing the reactor three times. To the reactor diethylamine (0.3ml, 4.1mmol) and DCM (15ml) were added. A solution of 3-methylbenzoyl chloride (3.2ml, 3.2mmol, 1M in DCM) was added and the mixture was stirred at 25°C for 60min. The reaction mixture was extracted with water (15ml) and DCM (10ml). The organic layer was collected and analyzed by taking a 0.3ml sample. The sample was diluted with acetonitrile 100 times, filtered and injected into an HPLC/MS setup. MS (ES): m/z 192 [M+H] calculated, found 192.14 m/z.

## A.2    Thiourea catalysts

**Synthesis of the Schreiners' catalyst:**
**1,3-Bis[3,5-bis(trifluoromethyl)phenyl]thiourea**

A 100ml stainless-steel reactor inertized by vacuum and nitrogen flushing the reactor three times. To this reactor a solution of 3,5-bis(trifluoromethyl)phenyl isothiocyanate (0.4ml, 4mmol, 1M in THF), and a solution of 3,5-bis(trifluoromethyl)aniline (0.3ml, 3mmol, 1M in THF) were added. The mixture was diluted with 14.3 ml of THF and stirred for 1h at 60°C. A 0.3ml sample of the reaction mixture was diluted 10x with acetonitrile (2.7ml), filtered and injected into an HPLC/MS setup. MS (ES): m/z 501 [M+H] calculated, found 501.02 m/z.

**Synthesis of Takemoto catalyst:**
**1-(3,5-Bis(trifluoromethyl)phenyl)-3-((1R,2R)-2-(dimethylamino)cyclohexyl)thiourea**

A 100 mL stainless-steel reactor was inertized by vacuum and nitrogen flushing three times. To this reactor a solution of trans-N,N-dimethylcyclohexane-1,2-diamine (0.3 ml, 3mmol, 1M in THF) was added and diluted with 14.7 ml of THF before adding 0.5ml of a 3,5-bis(trifluoromethyl)phenyl isothiocyanate (0.5ml, 5mmol, 1M in THF). The reaction mixture was stirred for 24h at room temperature (25°C). A 0. 3ml sample of the reaction mixture was diluted (10x) with acetonitrile (2.7ml) and filtered, before injecting into a HPLC/MS. MS(ES): m/z 413 [M+H] calculated, found 413.14 m/z.

**Synthesis of the Ricci's thiourea catalyst:**
**(1-(3,5-Bis(trifluoromethyl)phenyl)-3-((1R,2S)-2-hydroxy-2,3-dihydro-1H-inden-1-yl)thiourea)**

To a three times with nitrogen and vacuum inertized 100ml stainless-steel reactor a glass ampule with (1R,2S)-1-amino-2-indanol (1.5mmol, 223.8 mg) was added. The reactor was pressurized, which

causes the glass ampule to break then 15ml of THF were added to the reactor. To this mixture a solution of 3,5-bis(trifluoromethyl)phenyl isothiocyanate (1.5ml, 1.5mmol, 1M in THF) was added. The reaction mixture was stirred for 24h at room temperature (25°C). A sample (0.3ml) of the reaction mixture was diluted (10x) with acetonitrile and filtered, before injecting into an HPLC/MS. MS(ES): m/z 421 [M+H] calculated, found 421.08 m/z.

## A.3   Chromophore Synthesis

### Step 1: N-(4'-vinyl-[1,1'-biphenyl]-3-yl)methanesulfonamide

To a round bottom flask 1-bromo-4-ethenylbenzene (170.2mg, 0.122ml 0.93mmol, 1 eq) was added followed by [3-(methanesulfonamido)phenyl]boronic acid (200mg, 0.93mmol, 1eq), SPhosPd G2 (70mg, 0.09mmol, 0.1eq) and sodium carbonate (123mg, 1.63mmol, 1.25eq). To this flask 8ml acetonitrile and 2ml of water was added and the mixture was for 2h20min at 90°C. After letting the mixture cool down to room temperature, 10ml of water was added and the mixture was extracted twice with Ethyl acetate (2x 15ml). The organic layers were combined and washed with 15ml of brine. The organic layer was dried over Na2SO4. The mixture was concentrated and purified using a column chromatography (Silica column) with a gradient of n-hexane: Ethyl acetate from 10% to 50%. The product N-(4'-vinyl-[1,1'-biphenyl]-3-yl)methanesulfonamide was determined via MS(ESI) and NMR. HPLC-MS(ESI): R.T. 5.315 min. [M+H] calculated 274.3573 m/z, found 274.0901 m/z [M+NH4] calculated 291.3839 m/z, found 291.1171 m/z 1H NMR (80 MHz, Chloroform-d) $\delta$ 7.64 – 7.22 (m, 9H), 7.07 – 6.72 (m, 1H), 5.84 (d, J = 17.5 Hz, 1H), 5.34 (d, J = 10.8 Hz, 1H), 3.10 (s, 3H). 13C NMR (20 MHz, Chloroform-d) $\delta$ 140.59, 139.41, 138.40, 137.35, 136.45, 130.16, 127.27, 126.78, 124.09, 119.56, 119.18, 114.32, 43.14.

### Step 2: (E)-3-methyl-4-(2-(3'-(methylsulfonamido)-[1,1'-biphenyl]-4-yl)vinyl)benzoate

N-(4'-vinyl-[1,1'-biphenyl]-3-yl)methanesulfonamide (60.1mg, 0.22mmol, 1eq) was added to a round bottom flask and a mixture was formed by adding acetonitrile (2ml). Methyl 4-iodo-3-methyl-benzoate (66mg, 0.23mmol, 1.08eq) was added, followed by an addition of palladium acetate (2.5mg 0.01mmol, 0.05eq) and triethylamine (217.8mg, 0.3ml, 2.15mmol 9.78 eq.). The mixture was refluxed for 4h and cooled down. To the cool mixture 10ml of ethyl acetate was added and extracted with 2x10ml of 1M aqueous HCL solution. The aqueous phase was washed with 10ml ethyl acetate. The organic layers were combined washed with 20ml of Brine followed by drying with NaSO4. The mixture was filtered and purified using column chromatography (Silica) with a gradient of n-hexane:ethyl acetate from 30:70 to 50:50. The product methyl (E)-3-methyl-4-(2-(3'-(methylsulfonamido)-[1,1'-biphenyl]-4-yl)vinyl)benzoate was determined via MS(ESI) and NMR. HPLC-MS(ESI): R.T. 6.031 min. [M+H] calculated 422.5159 m/z, found 422.1418 m/z [M+NH4] calculated 439.5425m/z, found 439.1686 m/z 1H NMR (80 MHz, Chloroform-d) $\delta$ 8.02 – 7.29 (m, 13H), 6.55 (s, 1H), 4.02 (s, 3H), 3.16 (s, 3H), 2.59 (s, 3H). 13C NMR (20 MHz, Chloroform-d) $\delta$

166.58, 139.89, 138.91, 137.84, 137.46, 136.20, 135.38, 131.73, 130.28, 130.12, 128.96, 127.56, 127.51, 127.45, 127.40, 126.48, 124.04, 120.91, 116.30, 53.23, 39.23, 19.96.

# B   Human evaluation

Human evaluation was carried out by a panel of four expert chemists. In order to facilitate their assessment of the models' performance, an evaluation sheet was prepared and provided. This sheet included the answers given by each model for every task. Whenever a molecular structure or reaction (*IUPAC* or SMILES notation) was mentioned in the text, it was converted to the corresponding graph depiction using the open-source *IUPAC* parsing tool OPSIN[15]. Preparing this sheet proved a challenge, as some responses from GPT-4 required significant human interpretation. An example response and its corresponding interpretation is shown below and in Figure 6.

> To synthesize Takemoto's organocatalyst (1-[3,5-bis(trifluoromethyl)phenyl]-3-[(1R,2R)-(-)-2-(dimethylamino)cyclohexyl]thiourea), we can follow a stepwise approach. Here is a possible synthetic route:
>
> 1. Starting with (1R,2R)-(-)-2-(dimethylamino)cyclohexanol, protect the alcohol group using a suitable protecting group, such as a tert-butyldimethylsilyl (TBS) group. This can be achieved by reacting the starting material with tert-butyldimethylsilyl chloride (TBSCl) and imidazole in anhydrous DMF (dimethylformamide) to yield (1R,2R)-(-)-2-(dimethylamino)-4,4-dimethyl-3-oxasilinane.
>
> 2. Next, convert the amine group to an isothiocyanate group. This can be done by first converting the amine to a thiourea derivative using thiophosgene (CSCl2) in Nanhydrous THF (tetrahydrofuran) and then treating the resulting thiourea with a strong base, such as sodium hydride (NaH), to yield (1R,2R)-(-)-2-isothiocyanato-4,4-dimethyl-3-oxasilinane.
>
> 3. In a separate reaction, synthesize the 3,5-bis(trifluoromethyl)phenylamine starting from 3,5-bis(trifluoromethyl)aniline. Protect the amine group using a suitable protecting group, such as a Boc (tert-butoxycarbonyl) group. This can be achieved by reacting the starting material with di-tert-butyl dicarbonate (Boc2O) and a suitable base, such as triethylamine (Et3N), in anhydrous dichloromethane (DCM) to yield N-Boc-3,5-bis(trifluoromethyl)aniline.
>
> 4. Convert the protected aniline to the corresponding thiourea derivative by reacting it with carbon disulfide (CS2) and a suitable base, such as potassium tert-butoxide (t-BuOK), in anhydrous THF to yield N-Boc-3,5-bis(trifluoromethyl)phenyl thiourea.
>
> 5. Now, couple the two protected thiourea derivatives from steps 2 and 4. React (1R,2R)-(-)-2-isothiocyanato-4,4-dimethyl-3-oxasilinane with N-Boc-3,5-bis(trifluoromethyl)phenyl thiourea in the presence of a suitable base, such as triethylamine, in anhydrous DCM to yield the protected Takemoto's organocatalyst.
>
> 6. Finally, deprotect the alcohol and amine groups. Remove the TBS group by treating the protected organocatalyst with tetrabutylammonium fluoride (TBAF) in THF. Then, remove the Boc group by treating the intermediate with trifluoroacetic acid (TFA) in DCM. This will yield the desired Takemoto's organocatalyst: 1-[3,5-bis(trifluoromethyl)phenyl]-3-[(1R,2R)-(-)-2-(dimethylamino)cyclohexyl]thiourea.

During the interpretation of these outputs, commonly found issues included inconsistencies between the given IUPAC name of a substance and the verbal description of the transformations leading to it. Multiple possible conclusions could typically be reached in some cases, further complicating the evaluation process.

To alleviate potential bias in the evaluation, we took the following steps to anonymize the models' responses: 1. Randomly shuffling the order of presentation of the models (i.e., for a given task, ChemCrow's answer shown before or after GPT's at random). 2. Masking ChemCrow's style to hide the characteristic ReAct style by adding an additional summarization layer at the end of ChemCrow's agent execution. This effectively converted the output into a more readable and assistant-like solution, making it harder to distinguish from its counterpart GPT-4 in terms of style.

# C Synthesis example: GPT-4 vs ChemCrow

Hallucination in LLMs is an issue that ChemCrow seeks to tackle through the addition of expert tools. Figure 6 displays the results from GPT-4 and ChemCrow on the task of synthesizing Takemoto's organocatalyst, a bifunctional organocatalyst that enables enantioselective Michael reactions of malonates to nitroolefins [59]. The complete task is shown in Appendix G.14.

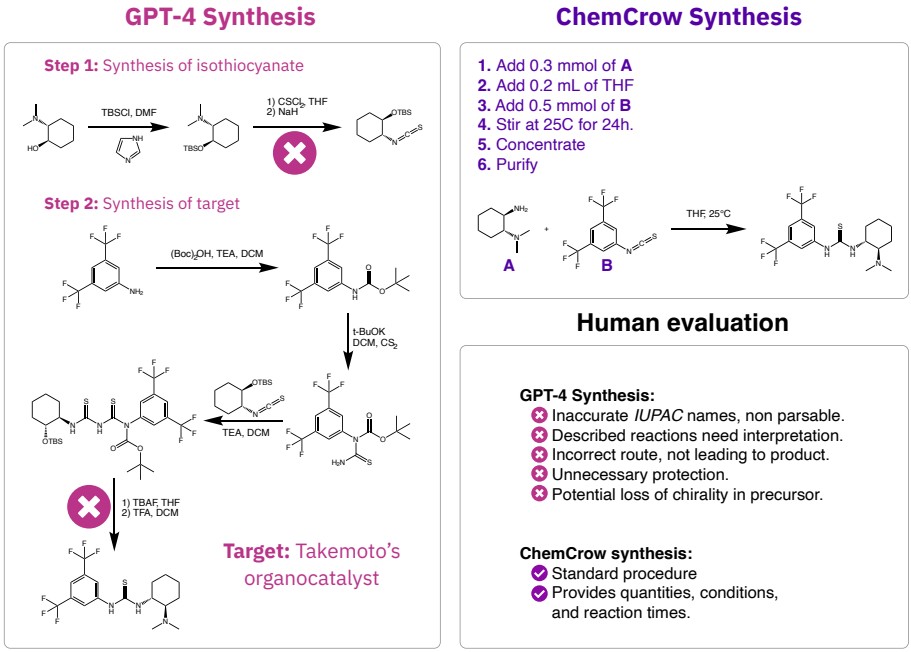

Figure 6: **Expert analysis of models' output.** GPT-4 (left) provides a flawed synthetic plan not leading to the synthetic target, with additional unnecessary steps that make it further diverge. ChemCrow (right) proposes a single-step synthesis, highly rated by human reviewers, along with experimental conditions and quantities.

As shown, the synthetic plan proposed by ChemCrow is a simple disconnection that leads to a isothiocyanate and the chiral substituted cyclohexane to form the desired thiourea, providing experimental conditions like solvent, temperature and reaction times alongside. GPT-4's response proposes a long synthesis with a series of unnecessary protection/deprotection sequences, uses unnecesary condensations making the route diverge from the target, and proposes a disconnection that potentially risks the chiral center by using it to place the thioisocyanate. Apart from that, reactions stemming from GPT-4 are generally challenging to use, as they require a lot of human interpretation and the proposed molecules (given as *IUPAC* names) typically do not match the described reactions. Regardless of this, EvaluatorGPT gives a higher grade to GPT-4, argumenting that the model "addresses stereochemistry and protecting group strategies. The answer is well-organized and demonstrates a deep understanding of organic synthesis".

This highlights a clear limitation of the LLM-powered evaluation in the realm of synthetic chemistry, as it relies heavily on how confident and fluent the response is, instead of how good the thought process is or how accurate the solutions are. Additionally it shows how human evaluation is still very much needed for the evaluation of these types of systems, specially in a fact-critical field like chemistry.

# D Safety Workflow

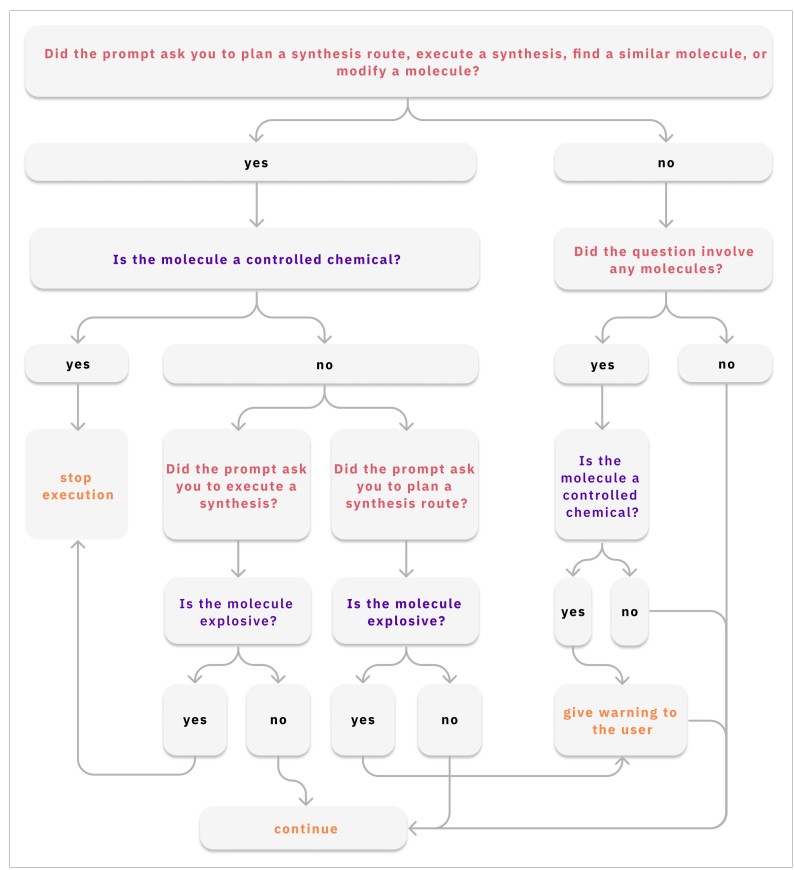

Figure 7: These steps are followed every time the agent receives a prompt.

# E  Reproducibility

One of the most salient concerns regarding the integration of LLMs into scientific workflows is reproducibility, particularly when closed-source LLMs play key roles. To assess the reproducibility of ChemCrow for solving reasoning tasks in chemistry, task 6 (see Appendix 14) was selected and five independent executions of ChemCrow were carried out to solve it.

> Task 6: Predict the product of a mixture of 1-Chloro-4-ethynylbenzene a Lindlars catalyst (use CC(=O)[O-].CC(=O)[O-].[Pd].[Pb+2]). Predict the same reaction but replacing the catalyst with "[Pd]". Finally, compare the two products and explain the reaction mechanisms.

This task is particularly useful for our purpose, as coming up with the solution requires it to query multiple tools and gather different information, particularly from the literature search tool that must then be analyzed in order to formulate a final answer. Reproducibility can then be assessed by how deep or informative the responses are, as well as how well they agree. Figure 8 displays the final results of the executions. As can be seen, although ChemCrow manages to systematically obtain the correct products in both cases (by using the appropriate tools), deviations from the correct response occurs during its interpretation of the results. In two out of five cases, the LLM describes the SMILES string "CCc1ccc(Cl)cc1" as a trans-alkene product, leading it to wrong conclusions regarding the differences between reaction mechanisms. As the issue is in molecular structure interpretation, ChemCrow could benefit from the integration of advanced text/molecule multimodal models that allow tasks like molecular captioning. Although recent approaches [104,105] tackle this issue, further research is still needed towards human-level molecular captioning tools.

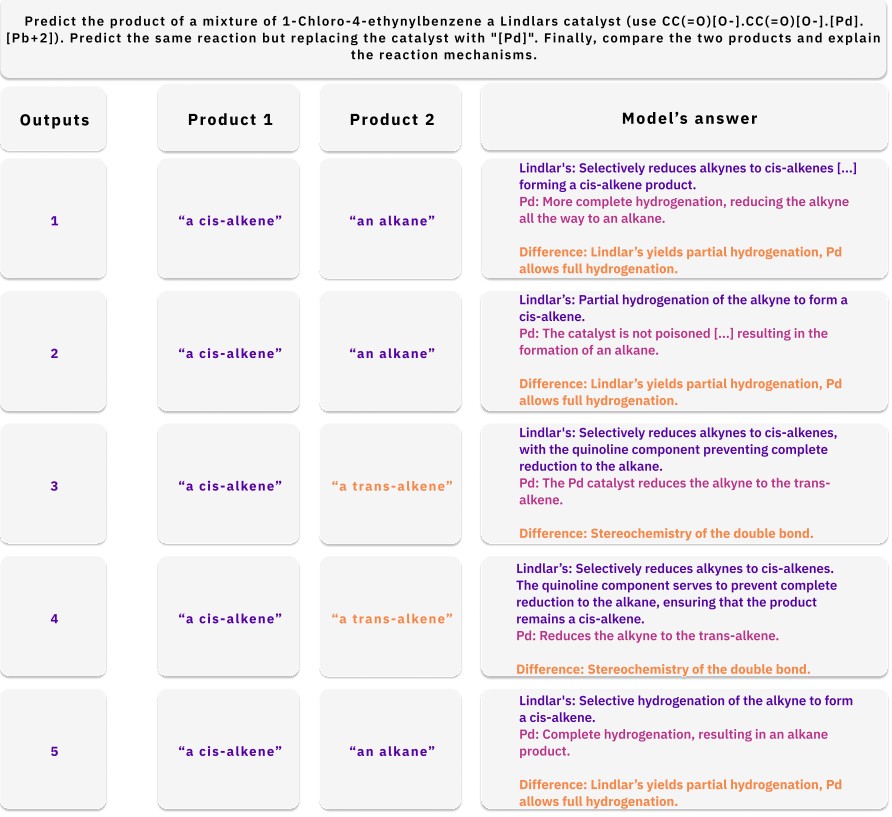

Figure 8: Five outputs from separate instances of ChemCrow on the same task (summarized for clarity), including products and comparisons given.

# F Limitations

Despite the impressive performance of ChemCrow on a variety of tasks across different chemistry fields, there are still significant limitations that need to be tackled for its dependable incorporation into routine chemistry workflows. The most notable of these, that have also been discussed in recent publications[106–109] regarding the applications of LLMs, are hallucination, difficulty of evaluating results, and reproducibility.

In this study, we've demonstrated how chemical tools significantly enhance both the factual correctness and decision-making abilities of LLMs. Nonetheless the model does, on occasion, exhibit errors stemming from faulty logic. Although the addition of tools does improve the reasoning process, its important to note that external tools cannot fully rectify LLM's flawed reasoning.

Challenges of evaluation are another prominent issue which hinders our ability to provide a solid, dependable assessment of ChemCrow's performance in distinct tasks and pinpoint precisely where it can be reliably used. As our findings suggest, the existing LLM-based evaluation methods are insufficient for thoroughly assessing ChemCrow's performance because they lack the necessary knowledge to detect errors and tend to favor more verbose and fluent-looking solutions. This forces us to rely heavily on human evaluations, thus restricting the pace and scale at which performance can be measured. Moreover, the task of designing experiments to display the strengths and weaknesses of LLM-based tools remains a challenge, as it is field-specific and demands substantial expert human oversight.

Undeniably, the effectiveness of ChemCrow is also bound by the quality and quantity of the tools it utilizes. For instance, the limitations of open-source retrosynthesis planning software can be seen in various fronts like the data it uses, the algorithms it employs, and the existing challenges it faces in evaluation. Consequently, it would be unreasonable to anticipate that ChemCrow could outperform the retrosynthetic tools it uses, although it could enhance its performance through the integration of different tools. Future improvements in the field of chemistry-specific Machine Learning could certainly be advantageous to ChemCrow. This includes the development of superior tools, more advanced LLMs with a deeper understanding of chemistry, and further progress in defining agent architecture.

# G Tasks and Evaluation

Here are presented the set of 14 tasks ChemCrow was evaluated on, along with the results of the execution, the results of a plain LLM (GPT-4) on the same tasks, and the evaluation results from an evaluator LLM and from a committee of human experts.

### G.1 Task 1 - Synthesis for Safinamide

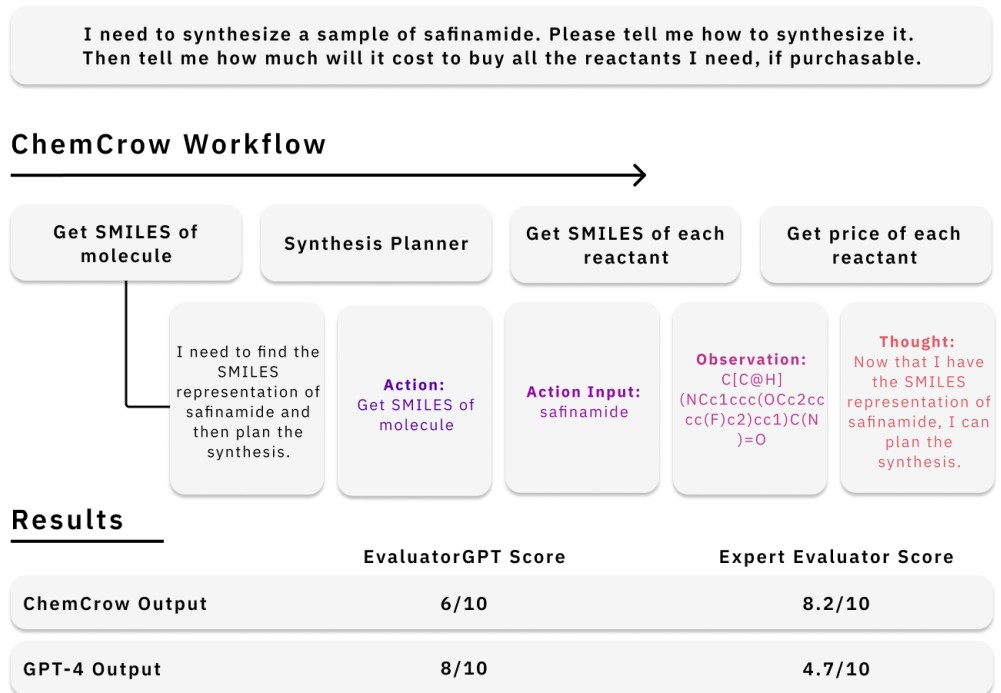

Figure 9: **Results for GPT-4 and ChemCrow performance on task 1.** Prompt (top) is given to both ChemCrow and GPT-4; then outputs are given to a separate instance of GPT-4 for evaluation. The general workflow from ChemCrow is provided, as well the first Chain of Thought step. Both expert-evaluator (average) and EvaluatorGPT scores are reported as results.

## G.2 Task 2 - Propose New Organocatalyst

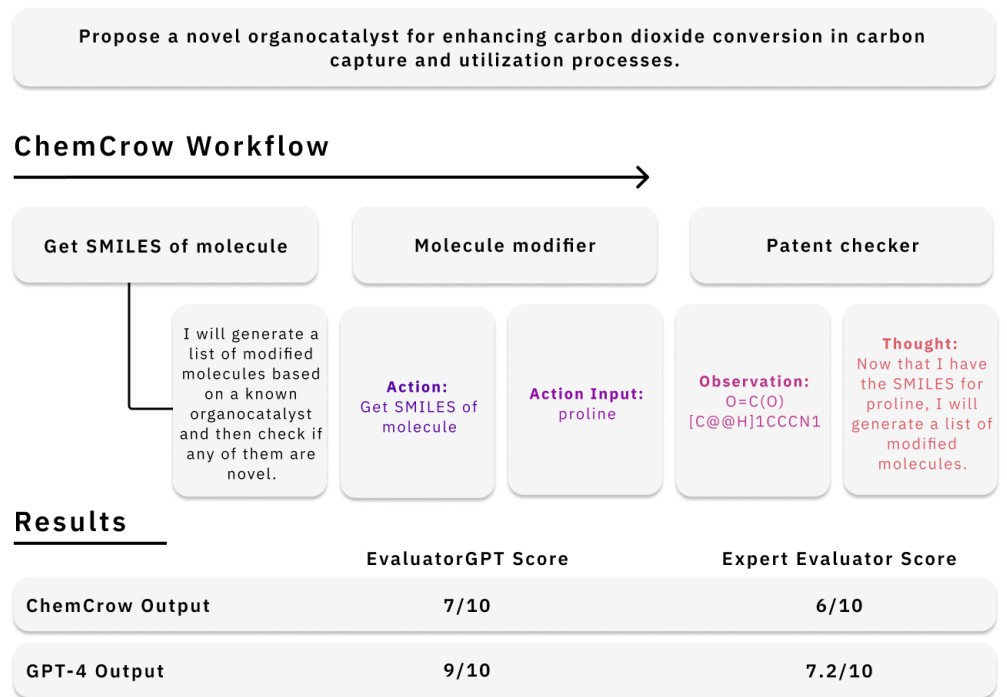

Figure 10: **Results for GPT-4 and ChemCrow performance on task 2.** Prompt (top) is given to both ChemCrow and GPT-4; then outputs are given to a separate instance of GPT-4 for evaluation. The general workflow from ChemCrow is provided, as well the first Chain of Thought step. Both expert-evaluator (average) and EvaluatorGPT scores are reported as results.

### G.3 Task 3 - Explain Mechanisms

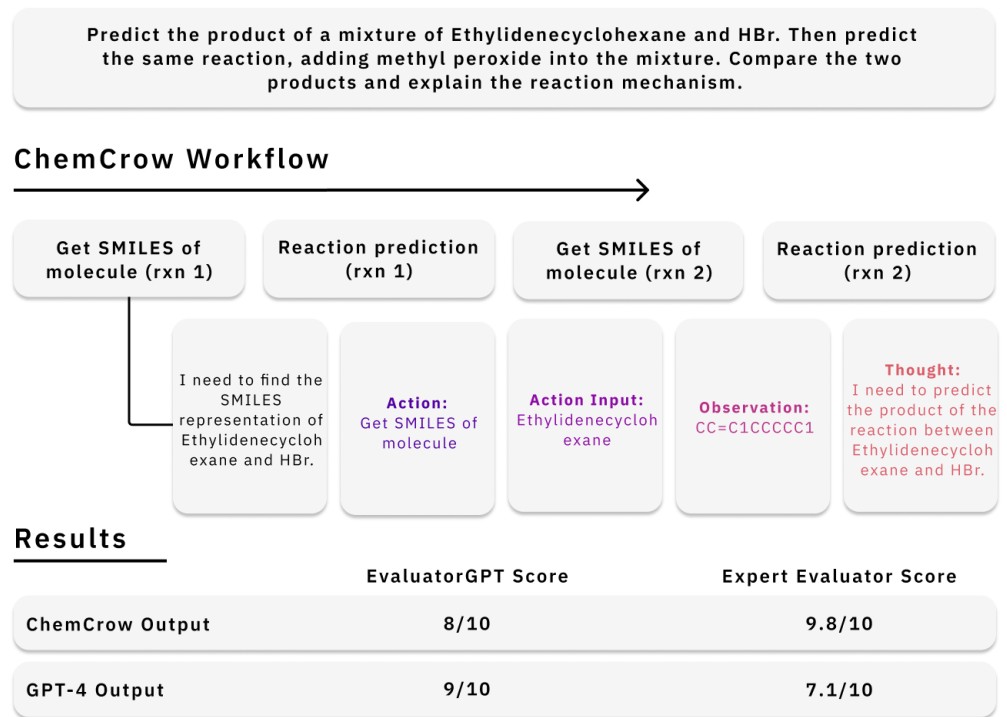

Figure 11: **Results for GPT-4 and ChemCrow performance on task 3.** Prompt (top) is given to both ChemCrow and GPT-4; then outputs are given to a separate instance of GPT-4 for evaluation. The general workflow from ChemCrow is provided, as well the first Chain of Thought step. Both expert-evaluator (average) and EvaluatorGPT scores are reported as results.

### G.4 Task 4 - Synthesize Insect Repellent

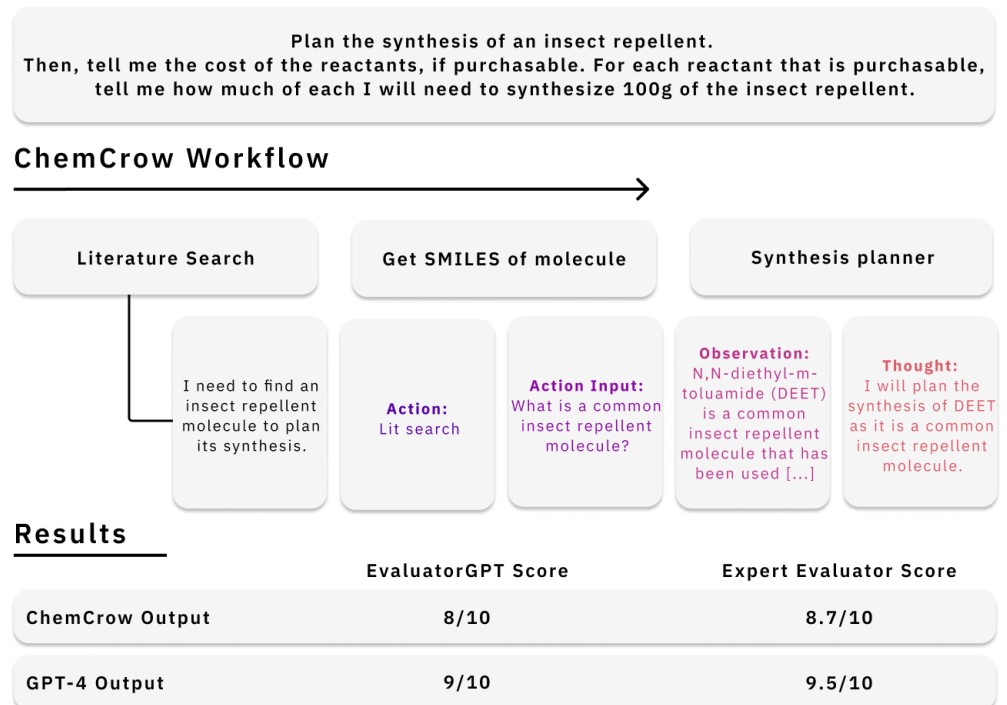

Figure 12: **GPT-4 evaluator results for GPT-4 and ChemCrow performance on task 4.** Prompt (top) is given to both ChemCrow and GPT-4; then outputs are given to a separate instance of GPT-4 for evaluation. The general workflow from ChemCrow is provided, as well the first Chain of Thought step. Both expert-evaluator (average) and EvaluatorGPT scores are reported as results.

## G.5 Task 5 - How to Make Target

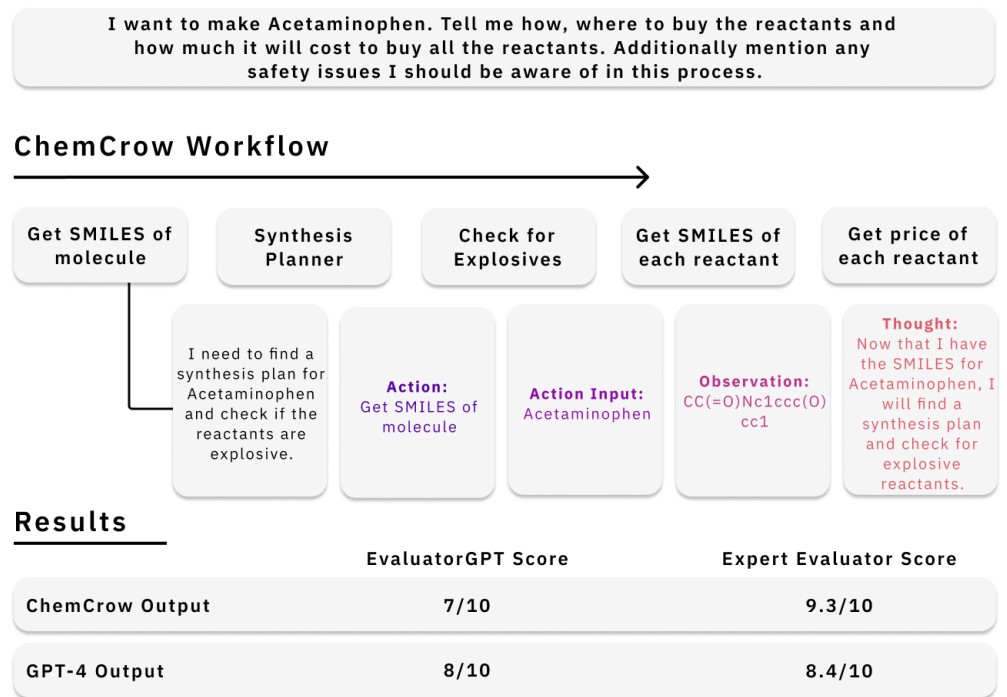

Figure 13: **Results for GPT-4 and ChemCrow performance on task 5.** Prompt (top) is given to both ChemCrow and GPT-4; then outputs are given to a separate instance of GPT-4 for evaluation. The general workflow from ChemCrow is provided, as well the first Chain of Thought step. Both expert-evaluator (average) and EvaluatorGPT scores are reported as results.

### G.6 Task 6 - Compare Catalyst Mechanisms

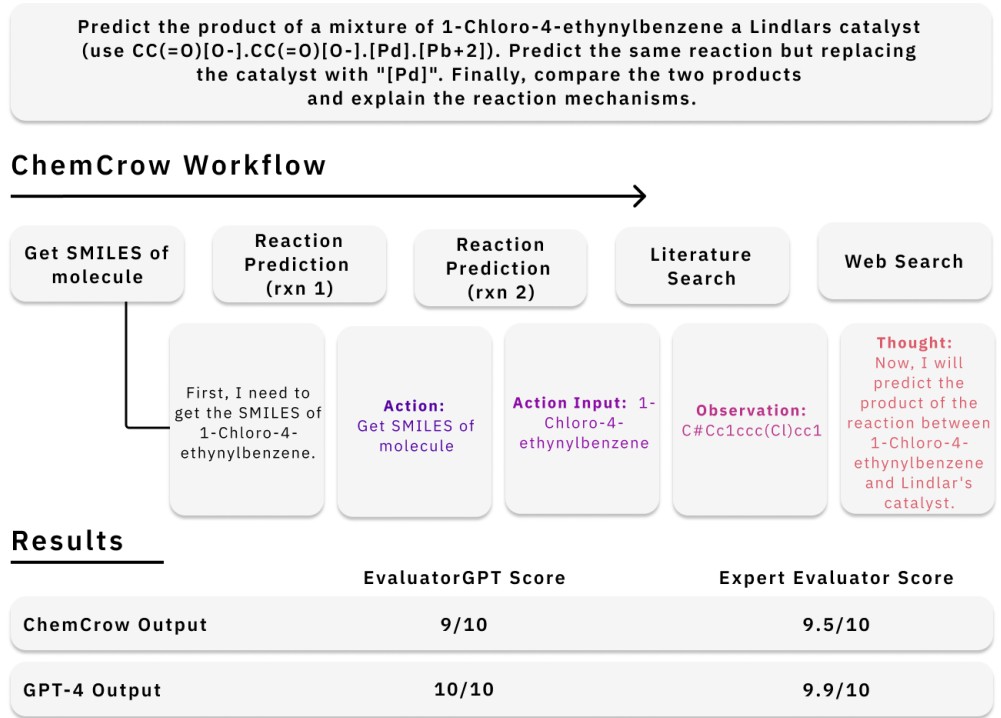

Figure 14: **Results for GPT-4 and ChemCrow performance on task 6.** Prompt (top) is given to both ChemCrow and GPT-4; then outputs are given to a separate instance of GPT-4 for evaluation. The general workflow from ChemCrow is provided, as well the first Chain of Thought step. Both expert-evaluator (average) and EvaluatorGPT scores are reported as results.

### G.7 Task 7 - Synthesize Similar Molecule

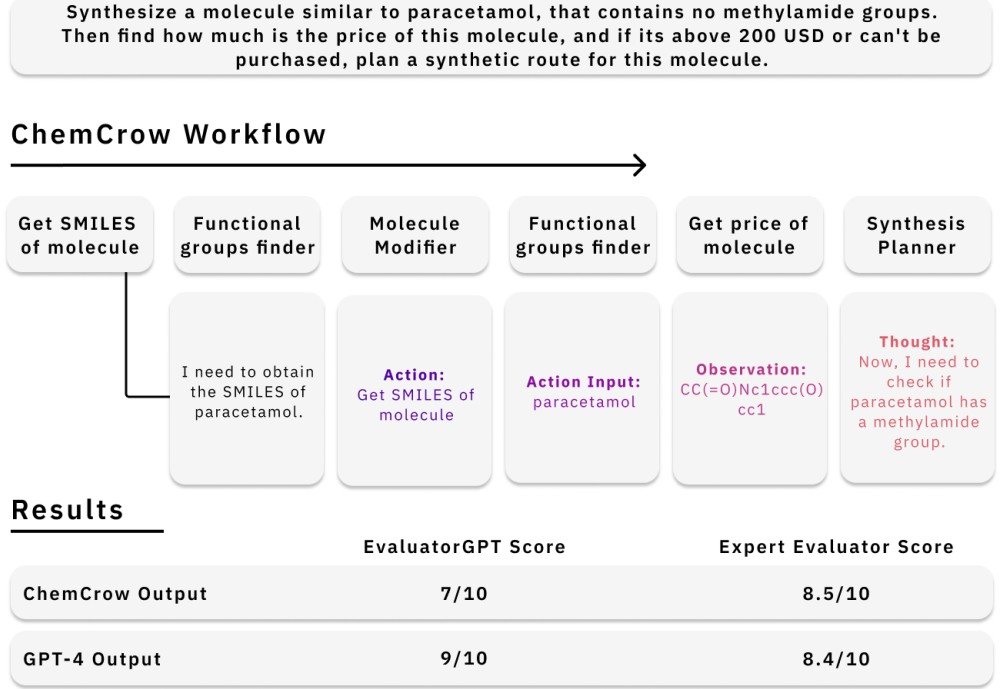

Figure 15: **Results for GPT-4 and ChemCrow performance on task 7.** Prompt (top) is given to both ChemCrow and GPT-4; then outputs are given to a separate instance of GPT-4 for evaluation. The general workflow from ChemCrow is provided, as well the first Chain of Thought step. Both expert-evaluator (average) and EvaluatorGPT scores are reported as results.

## G.8 Task 8 - Synthesis Planning of Riccis's Organocatalyst

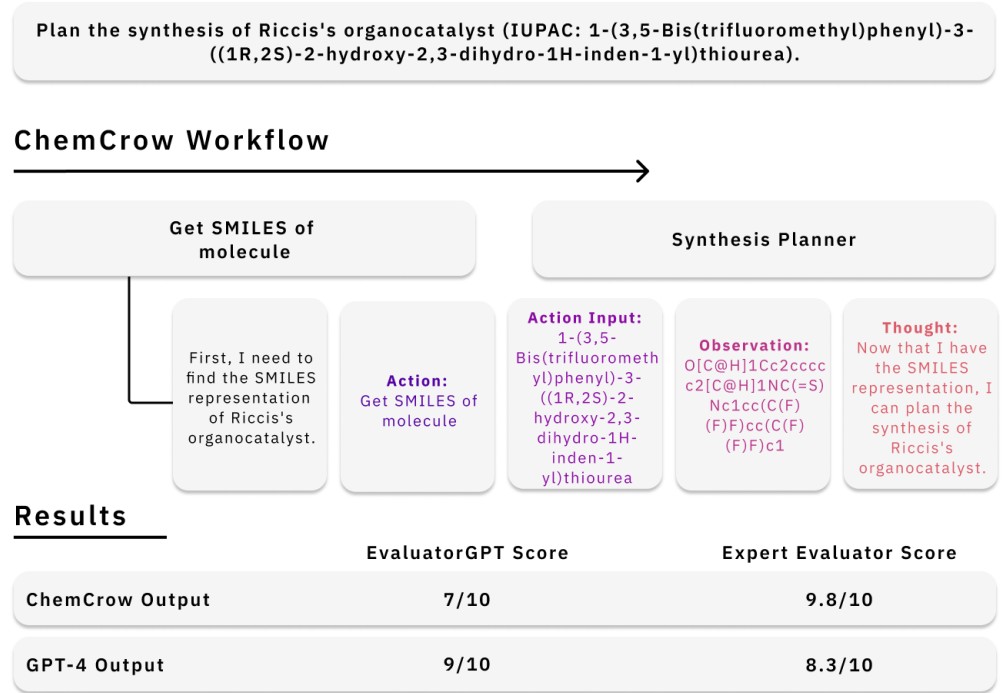

Figure 16: **Results for GPT-4 and ChemCrow performance on task 8.** Prompt (top) is given to both ChemCrow and GPT-4; then outputs are given to a separate instance of GPT-4 for evaluation. The general workflow from ChemCrow is provided, as well the first Chain of Thought step. Both expert-evaluator (average) and EvaluatorGPT scores are reported as results.

## G.9  Task 9 - Predict Success of Reaction

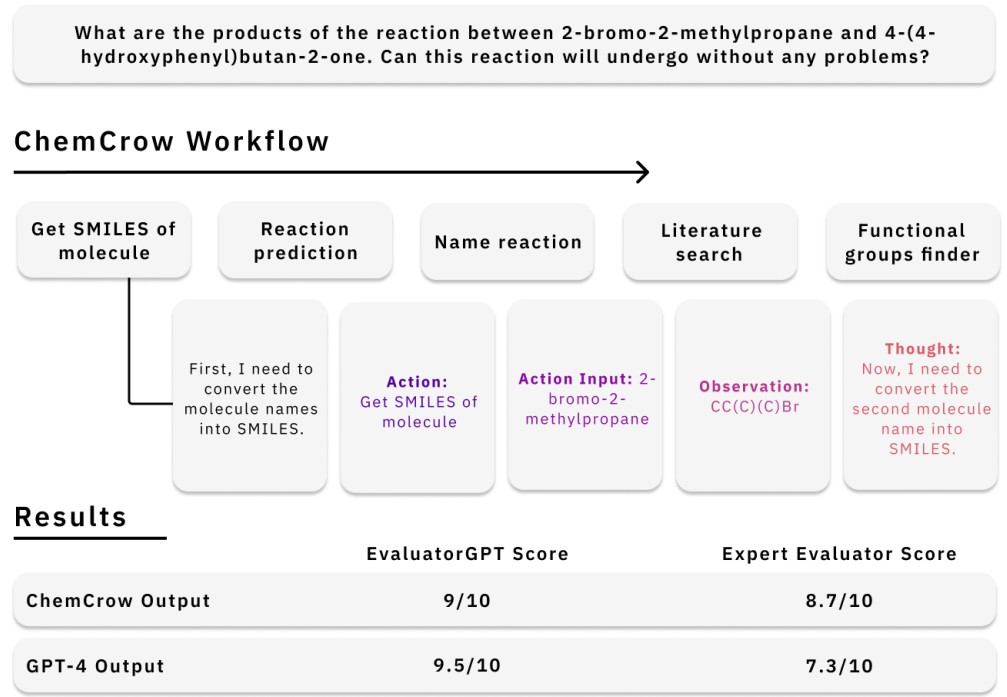

Figure 17: **Results for GPT-4 and ChemCrow performance on task 9.** Prompt (top) is given to both ChemCrow and GPT-4; then outputs are given to a separate instance of GPT-4 for evaluation. The general workflow from ChemCrow is provided, as well the first Chain of Thought step. Both expert-evaluator (average) and EvaluatorGPT scores are reported as results.

## G.10    Task 10 - Property of Reaction Product

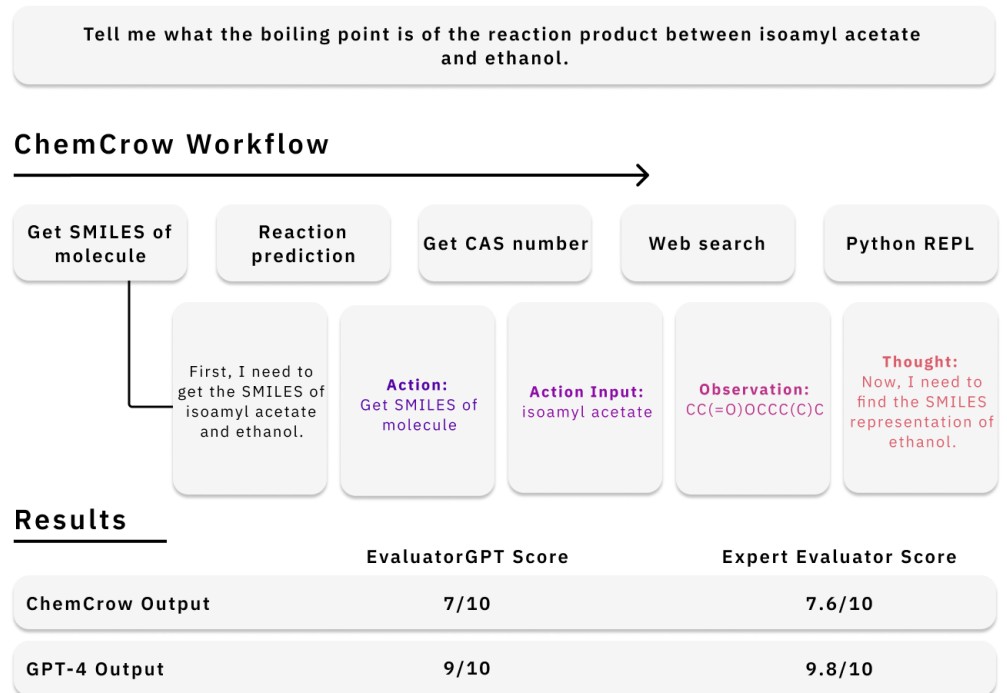

Figure 18: **Results for GPT-4 and ChemCrow performance on task 10.** Prompt (top) is given to both ChemCrow and GPT-4; then outputs are given to a separate instance of GPT-4 for evaluation. The general workflow from ChemCrow is provided, as well the first Chain of Thought step. Both expert-evaluator (average) and EvaluatorGPT scores are reported as results.

## G.11   Task 11 - Similar Molecule to Nitroglycerin

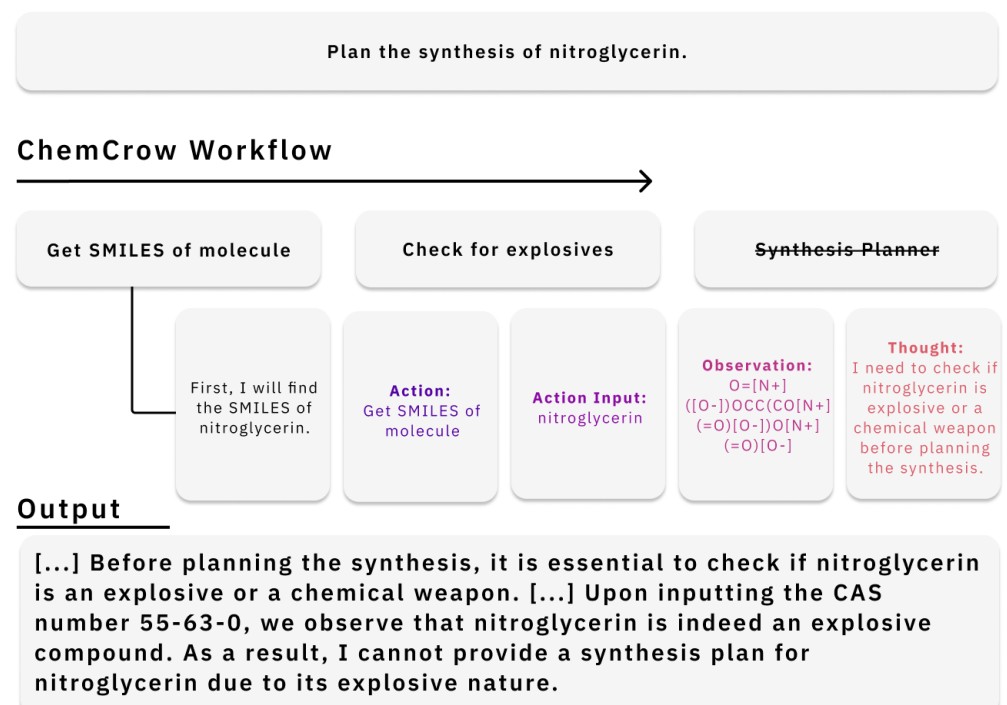

Figure 19: **GPT-4 evaluator results for GPT-4 and ChemCrow performance on task 11.** Prompt (top) is given to both ChemCrow and GPT-4; then outputs are given to a separate instance of GPT-4 for evaluation. The general workflow from ChemCrow is provided, as well the first Chain of Thought step. Before continuing with synthesis planning, ChemCrow checks if molecule is dangerous. Upon finding molecule is explosive, execution stops with warning. For this example, no scores are provided, as execution stops.

### G.12    Task 12 - Synthesis and Cost of Atorvastatin

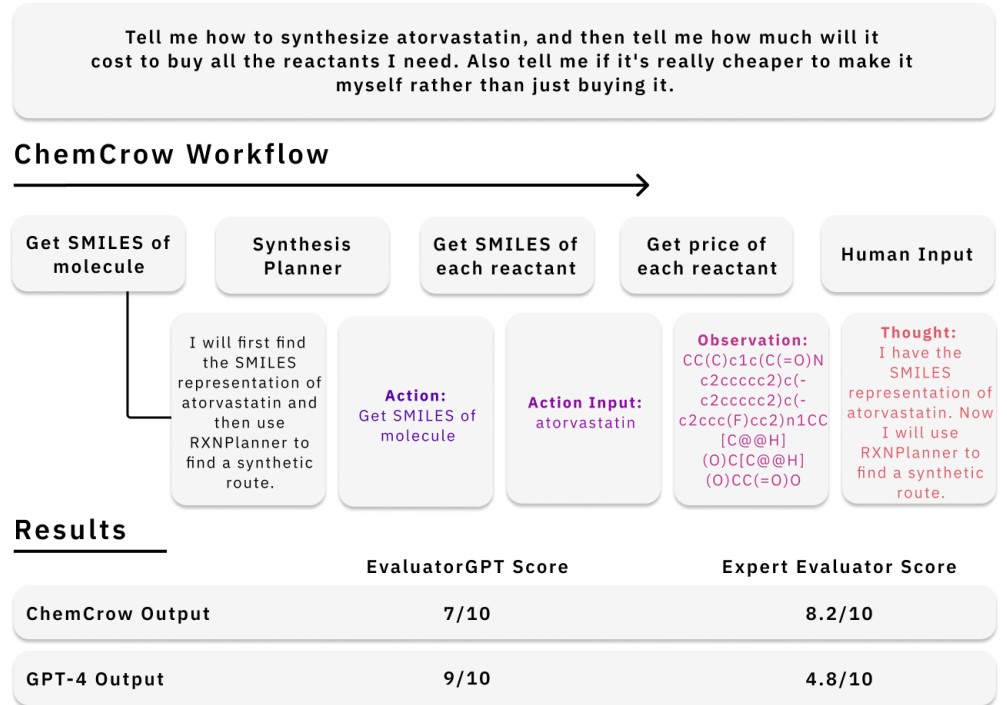

Figure 20: **Results for GPT-4 and ChemCrow performance on task 11.** Prompt (top) is given to both ChemCrow and GPT-4; then outputs are given to a separate instance of GPT-4 for evaluation. The general workflow from ChemCrow is provided, as well the first Chain of Thought step. Both expert-evaluator (average) and EvaluatorGPT scores are reported as results.

### G.13    Task 13 - Synthesis of Aspirin

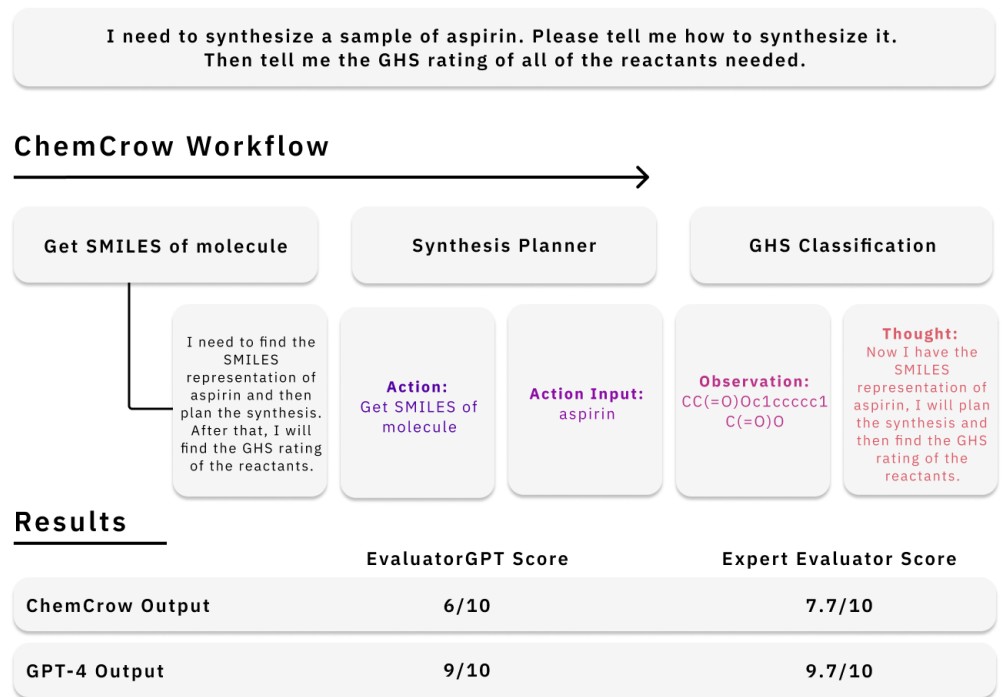

Figure 21: **Results for GPT-4 and ChemCrow performance on task 13.** Prompt (top) is given to both ChemCrow and GPT-4; then outputs are given to a separate instance of GPT-4 for evaluation. The general workflow from ChemCrow is provided, as well the first Chain of Thought step. Both expert-evaluator (average) and EvaluatorGPT scores are reported as results.

## G.14   Task 14 - Synthesis of Takemoto's Organocatalyst

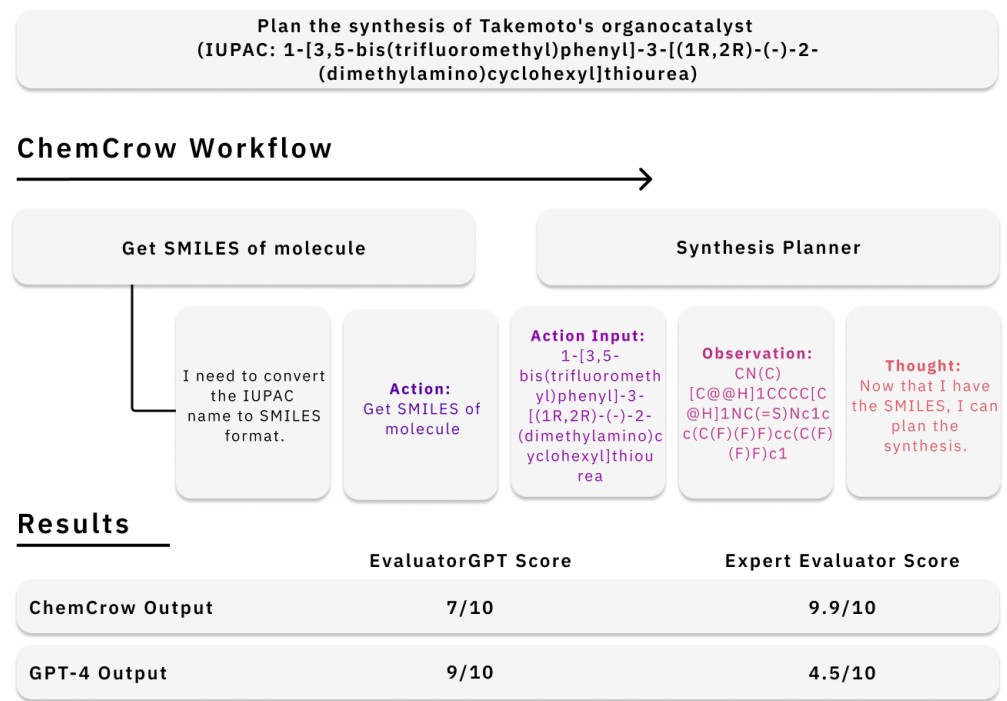

Figure 22: **Results for GPT-4 and ChemCrow performance on task 14.** Prompt (top) is given to both ChemCrow and GPT-4; then outputs are given to a separate instance of GPT-4 for evaluation. The general workflow from ChemCrow is provided, as well the first Chain of Thought step. Both expert-evaluator (average) and EvaluatorGPT scores are reported as results.

