# OpenReview forum: "Augmenting large language models with chemistry tools"
_NeurIPS.cc/2023/Workshop/AI4Science — NeurIPS2023-AI4Science Poster_

### Official Review · Reviewer_Pv6K · 2023-10-22
**Very interesting application paper on language agents and chemistry**

**Rating:** 9
**Confidence:** 4

**Review:**

**General comments**

This paper presents ChemCrow, an LLM agent designed for chemistry applications such as organic synthesis and material design. The authors augment existing LLMs (GPT-4) with ReACT prompting on 18 expert-designed tools. Both qualitative evaluation of case studies and quantitative evaluation on self-curated benchmarks have been performed, and the results show the effectiveness of ChemCrow. I especially appreciate that the authors also present risk mitigation strategies implemented by tools for responsible usage. Overall, this is an interesting paper that is well-presented. The authors have also made the tools and datasets publicly available, which can be valuable resources as well.

**Specific comments**

1. The authors might consider moving the methods part before results, or at least give some methodology descriptions (e.g., pseudo-code, flowcharts) before directly describing results.
2. The authors found that GPT-4-eval might not be able to distinguish clearly wrong completions. Maybe it is better to formulate the chemical tasks as multi-choice, which can be easily evaluated. The authors might also consider augmenting EvaluatorGPT with chemical tools (just for validation).
3. Are all ChemCrow experiments based on GPT-4? How about other LLMs, e.g. GPT-3.5?
4. Figure 4 color coding is confusing: I understand that ChemCrow is purple while GPT-4 is pink, but why in Figure 4d these colors are also used to code the pros & cons?
5. The authors might also discuss tool-augmented LLMs for other scientific domains, e.g. GeneGPT for biology.

---

### Official Review · Reviewer_wnR7 · 2023-10-23
**I think it is a nice paper elucidating the potential applications of LLM in chemistry. I am very positive to its acceptance**

**Rating:** 9
**Confidence:** 5

**Review:**

The paper demonstrates a commendable effort to leverage the capabilities of Large Language Models (LLM) for chemistry-related tasks.  The authors have presented a promising LLM chemistry agent, ChemCrow, that exhibits enhanced performance in specific chemical domains. Even though ChemCrow may not excel uniformly across all tasks, the authors' analysis of its limitations and proposed solutions adds depth and value to the paper. The significance of ChemCrow lies not only in its immediate utility but also in its potential to illustrate how cutting-edge AI technologies like LLM can revolutionize research in chemistry. The paper's broad view of both the advantages and challenges faced by LLM in chemical contexts makes it a compelling read, and I support its acceptance.

Questions and comments:

1. Could the authors clarify which specific version or configuration of GPT-4 was utilized in this study? This detail is essential to contextualize the results and understand any limitations associated with the chosen model.

2. For autonomous synthesis planning, it's essential to discern the success rate of ChemCrow's attempts. Did the agent manage successful syntheses in its first attempt, or were multiple iterations required? Additionally, if any human intervention or subsequent optimization was involved in the process, it should be transparently highlighted.

3. Recognizing the diversity of computational tools that chemists employ, is there a provision for users to integrate or plug in their preferred tools via an API with ChemCrow? This could enhance the platform's adaptability and relevance to a broader user base.

4. Relying on prompting for safety evaluation raises questions about the reproducibility and consistency of LLM responses. Given the critical nature of safety, would the authors consider administering repetitive safety challenges to gauge the model's success rate? An alternative approach might involve integrating hard-coded safety protocols to ensure consistent adherence to established safety standards.

---

### Meta-Review · Area_Chair_VTNS · 2023-10-26

**Recommendation:** Accept (Poster)
**Confidence:** 5

**Metareview:**

The paper explores augmenting LLMs with tools to solve chemistry related tasks. It is a timely topic, and Authors present a convincing case that adding tools is necessary to build robust and capable agents. Reviewers appreciated the technical soudness, novelty of the work, and the scope of the experiments. It is my pleasure to recommend the acceptance of the paper.